# Prediction of Potential Natural Antibiotics Plants Based on Jamu Formula Using Random Forest Classifier

**DOI:** 10.3390/antibiotics11091199

**Published:** 2022-09-05

**Authors:** Ahmad Kamal Nasution, Sony Hartono Wijaya, Pei Gao, Rumman Mahfujul Islam, Ming Huang, Naoaki Ono, Shigehiko Kanaya, Md. Altaf-Ul-Amin

**Affiliations:** 1Computational Systems Biology Lab, Graduate School of Science and Technology, Nara Institute of Science and Technology, Nara 630-0101, Japan; 2Department of Computer Science, Faculty of Mathematics and Natural Sciences, IPB University, Bogor 16680, Indonesia

**Keywords:** herbal plants, Jamu, natural antibiotics, prediction, Random Forest

## Abstract

Jamu is the traditional Indonesian herbal medicine system that is considered to have many benefits such as serving as a cure for diseases or maintaining sound health. A Jamu medicine is generally made from a mixture of several herbs. Natural antibiotics can provide a way to handle the problem of antibiotic resistance. This research aims to discover the potential of herbal plants as natural antibiotic candidates based on a machine learning approach. Our input data consists of a list of herbal formulas with plants as their constituents. The target class corresponds to bacterial diseases that can be cured by herbal formulas. The best model has been observed by implementing the Random Forest (RF) algorithm. For 10-fold cross-validations, the maximum accuracy, recall, and precision are 91.10%, 91.10%, and 90.54% with standard deviations 1.05, 1.05, and 1.48, respectively, which imply that the model obtained is good and robust. This study has shown that 14 plants can be potentially used as natural antibiotic candidates. Furthermore, according to scientific journals, 10 of the 14 selected plants have direct or indirect antibacterial activity.

## 1. Introduction

Jamu is the common name for traditional Indonesian medicines. Jamu medicines are prepared from plant roots, leaves, and any other parts of medicinal plants. Any specific herbal medicine is made from the combination of several types of plants considered to have efficacy [1]. Jamu medicines are used not only as a remedy for various diseases but also for health maintenance. These medicines have been used for centuries by the people of Indonesia for illness treatment. According to Indonesia’s National Food and Drug Agency, Jamu is traditional medicine created from a mixture of herbal plants whose formulations are usually passed down from generation to generation. Jamu development is still advantageous considering the abundant number of herbal plants available in Indonesia. It has been reported that Camellia sinensis could act against drug-resistant bacteria, e.g., MRSA and *P. aeruginosa* [2]. Therefore, Jamu formulas can be intuitively utilized for finding natural antibiotic plants.

Superbugs are bacteria that can fight drugs or antibiotics, e.g., *Staphylococcus aureus* resistant to methicillin [3]. This phenomenon is very worrying considering that discovering new antibiotics is not easy because it takes lots of money and time. If this problem is not immediately and adequately addressed, the number of deaths caused by antibiotic resistance is predicted to reach 10 million annually by 2050 [4]. In addition, persister and viable but non-culturable (VBNC) is also a medical challenge. Unfortunately, this subpopulation can withstand multidrug exposure via many putative molecular mechanisms [5]. Some research tries to address the VBNC problem, such as [6], which showed *lactate dehydrogenase* (RIMD2210633:ΔlldD) is thought to have a relationship with the occurrence of the VBNC state because deletion of this gene causes the cell to enter the VBNC stage. Protein aggregation is another factor that drives the persistence stage to the VBNC shift [7]. Knowledge of the phenotype of microbial cells is also crucial to determining the resistance of microbial pathogens. It is stated [8] that the fast-growing phenotypic variant avoids macrolide accumulation and survives antibiotic treatment without any gene mutations. Another study [9] tried to find the physiology of VBNC using microfluidics and time-lapse microscopy methods. The results show that VBNC cells are not dead or dying but share similar phenotypic features with persister cells.

The increase in antibiotic-resistant cases impacts the health sector and the economy. According to the European Center for Diseases Prevention and Control (ECDC), around 33,000 people die annually due to antibiotic-resistant problems [10]. Epidemiologists say the economic impact caused by antibiotic resistance is very significant. In America and some countries, it is stated that there are 11 million additional hospitalizations and more than USD 20 billion in additional treatment costs due to superbug or antibiotic-resistant problems [11]. Thailand also spends USD 203 million on antibiotics annually; however, many non-prescription antimicrobials are still used throughout the country, potentially increasing antimicrobial resistance [12].

Advances in artificial intelligence technology can now be used to accelerate the discovery of new antibiotics, predict antimicrobial resistance, and preliminary screening of novel antibiotic candidates. In 2020 in silico and in vivo approaches were combined to find new antibiotics [13]. This study used a deep neural network to predict molecules with antibacterial activity using various database sources such as drug repurposing hub and ZINC15. This study found eight antibiotic compounds (ZINC000098210492, ZINC000001735150, ZINC000225434673, ZINC000004481415, ZINC000019771150, ZINC000004623615, ZINC000238901709, and ZINC000100032716) with mostly different structures compared to known antibiotics. Gram stain data, site of infection, and patient demographics were utilized to build decision tools for determining antimicrobial resistance using eight machine learning methods. Another study [14] tried to combine machine learning with spectroscopy to predict the mechanism of action of specific antibiotics. Other work [15] could predict different mechanisms of action of antibiotics of the same class. The use of machine learning for research related to antibiotics, specifically Random Forest, has also been carried out by [16]. This study used a Random Forest to determine the relationship between bacterial phenotypic fingerprints and the mechanism of action of different compounds. Research conducted in vitro and in silico to search for antibiotics for class β-Lactam antibiotics [17] showed that Dihydroisocoumarin compounds isolated from the Wadi Lajab sediment fungus *Penicillium chrysogenum* has antimicrobial activity.

Antibiotics are one of the necessary components in treating diseases originating from bacterial infections. Antibiotics themselves are usually created from microorganisms that are toxic to other microorganisms (bacteria). In addition, plants can also produce compounds that inhibit bacterial growth (bacteriostatic). However, despite the effectiveness of antibiotics in treating diseases caused by bacterial infections, their extensive use has resulted in antibiotic resistance. The natural antibiotic approach is expected to reduce the problem of antibiotic resistance [18]. Natural antibiotics also have some advantages both for the user and the environment. The use of herbal ingredients has fewer side effects. In particular, the herbal component has a multi-function ability to treat several diseases at once [1]. On top of that, herbal ingredients are better for the surrounding environment because they require less industrial processing and necessitate growing more plants. This study uses a machine learning approach to find natural antibiotics based on herbal (Jamu) formulas at the plant level.

## 2. Results

### 2.1. Preliminary Screening Using Several Machine Learning Methods

We collected Jamu formulas from the KNApSAcK database from which we selected formulas effective against diseases instigated by bacteria and also diseases that are not caused by bacteria. We labeled the selected formulas as bacterial and non-bacterial. Our objective is to develop a robust model that can effectively classify the selected formulas into two classes. The important variables (plants in this case) attributing to a good model can then be utilized for identifying antibacterial plants. The KNApSAcK database (DB) contains information on the species–metabolite relationship (101.500), encompassing 20,741 species and 50,048 metabolites. This database also contains information on accurate mass, molecular formula, metabolite name, and mass spectra in several ionization modes. In addition, the Knapsack Family database contains information on traditional medicine (Kampo and Jamu), Kampo DB consists of 336 formulas with 278 medicinal plants, and Jamu consists of 5310 formulas with 278 medicinal plants [19].

As a preliminary test to determine the best prediction model for the Jamu formula dataset, we applied the lazypredict method of the *Scikit-learn* package. The results of the precursory screening can be seen in Table 1. Preliminary results using the lazypredict implied that the data would be better analyzed using the Random Forest (RF) method. Table 1 shows the results of various types of machine learning methods such as decision-tree-based (Random Forest, extra tree), kernel-based (Linear SVC, and NuSVC), distance-based (KNeighborsClassifier, and NearestCentroid), and probability-based (BernouliNB). The results from Table 1 show that the RF technique is the best classification model for the Jamu formula data with the highest values for accuracy, ROC-AUC, and F1-score compared to other methods.

The data used for this study are the herbal medicine formulas in terms of plants as constituents. Therefore, the formulas are the objects and the plants are the features in this case and class labels are types of diseases that can be treated by herbal medicines. For applying machine learning algorithms, the data were pre-processed to form a binary matrix in the form of [Jamu formula × plants] and two class labels for the diseases were assigned: bacterial and non-bacterial. The model performance can be improved by appropriate tuning of the model parameters.

### 2.2. Tuning Model Parameters for Random Forest

Parameter tuning is the process of determining the best parameters corresponding to a model. Hyperparameter tuning in Random Forest has been executed through 100 iterations using a grid search process. The tuned parameters correspond to the RF tool under the *Scikit-learn* library with the name *“sklearn.ensemble.RandomForestClassifier”*. Six parameters are considered for this study. *n_estimator* is the number of trees formed. Choosing a large number of *n_estimator* results in increased computational complexity. The maximum features used for modeling are selected by *max_features* while *max_depth* denotes the longest path between the root node and a leaf node to prevent the Random Forest from overfitting; *min_samples_split* is the parameter that minimizes the observations required at each node to divide it and *min_samples_split* with a value of five means that if any terminal node has more than five observations, it can be further divided into sub-nodes. In short, *min_samples_split* and *min_samples_leaf* make the distinction between leaf nodes and internal nodes. Bootstrap is a data sampling process in tree formation; if *‘false’*, then all data are used for sampling; if *‘true’*, then a data sampling process is carried out. The values used for tuning the parameters in this study can be seen in Table 2 and the best parameters obtained are as follows: *{‘n_estimators’: 1000, ‘min_samples_split’: 2, ‘min_samples_leaf’: 1, ‘max_features’: ‘sqrt’, ‘max_depth’: 110, ‘bootstrap’: True}*.

Out of 10 cross validations, the best accuracy, recall, and precision are 91.1%, 91.1%, and 90.0%, respectively. The detailed results can be seen in Table 3, and the ROC (receiver operating characteristic) curve for assessing the model performance can be seen in Figure 1. Here, Table 3 shows the metrics scores of each fold in terms of accuracy, recall, and precision. This result can be regarded as robust because the difference in values between folds is not more than 5 percent. The performance of the model in the best fold is displayed in Figure 1, which indicates that the model is quite good as the curve tends to be in the upper left region. The value of AUC (area under the curve) is approximately 92%.

### 2.3. Identification and Validation of Important Plants

Potential plants effective against bacterial diseases have been obtained by Random Forest algorithm based on the variable importance by using package *permutation_importance* under *Scikit-learn* library with threshold > 0. This criterion selected 14 important features that are considered potential candidates for natural antibiotic plants. The list of these plants is shown in Table 4. To validate our results, we searched the literature to find whether these plants can be used as antibiotics or to inhibit bacterial growth.

**Table 4 antibiotics-11-01199-t004:** Summary of the predicted plants.

Name of Plant	Habitat	Pharmacological Activities	References
*Clerodendrom squamatum*	Indonesia	*Staphylococcus aureus*, *Escherichia coli* and *Salmonella typhi bacteria*	[20,21]
*Prunus cerasus*	United States of America, Turkey, Russia, Serbia, Hungary, Iran, Austria, Azerbaijan, Germany, and Indonesia	Antibacterial activity	[22]
*Borreria hispida*	Indonesia	*Bacillus subtilis*, *Bacillus cereus*, *Staphylococcus aureus*, *Pseudomonas aeruginosa* and *Escherichia coli*	[23]
*Coptis chinensis*	China	*Escherichia coli*	[24,25]
*Cassia alata*	Indonesia	*Dermathophilus congolensis*, *Staphylococcus aureus*, *Corynebacterium parvum*, *Actinomyces bovis*, and *Clostridium septicum*	[26]
*Brucea javanica*	Indonesia	*Streptococcus pyogenes*	[27]
*Aglaia odorata*	Indonesia and China	*Bacillus cereus* ATCC 11778, *Staphylococcus aureus* ATCC 25923, *Acinetobacter baumannii* ATCC 19606 and *Escherichia coli* ATCC 25922	[28]
*Costus speciosus*	Indonesia	Antibacterial, antifungal, anticholinesterase, antioxidant, antihyperglycemic, anti-inflammatory, analgesic, antipyretic, antidiuretic, larvicidal, anti-stress and estrogenic activity	[29]
*Stachytarpheta jamaicensis*	Indonesia	*Bacillus subtilis*, *Escherichia coli*, *Staphylococcus aureus*, *Pseudomonas aeruginosa*, *Proteus vulgaris*, *Klebsiella aerogenes*, *Proteus mirabilis**and Candida albicans.*	[30,31]
*Trichosanthes kirilowii*	China	*Bacillus cereus*, *Escherichia coli*, and *Streptococcus faecalis*.	[32]
*Prunus armeniaca* L.	US, Turkey, and Indonesia	Antimicrobial, antimutagenic, inhibiting enzymes, cardioprotective, anti-inflammatory and antinociceptive	[33]
*Fritillariae cirrhosae bulbus*	China	Antitussive, expectorant, analgesic, anti-cancer, anti-inflammatory, and antioxidative.	[34]
*Scaphium affinis*	Indonesia	Used to treat acute cough, sore throat, hemorrhoids, and increase female fertility	-
*Pueraria lobata*	China	Antioxidant, antiglycation, skin generation, and melanogenesis	[35]

Out of 14 predicted plants, 10 were found to be directly or indirectly used as antibiotics, antibacterial, and general bacterial inhibitors according to various sources. The validation process adopted in this work uses scientific journals and publicly available databases (KNApSAcK and TCM). Below we describe 10 validated plants.

*Clerodendrom squamatum* or better known as sesewanua leaf by the people of North Sulawesi, Indonesia, has often been used as a traditional medicine to treat fever, fractures, and swelling [18]. As stated by [19], sesewanua leaf extract using 96% ethanol by the Kirby and Bauer diffusion method could inhibit the growth of *Staphylococcus aureus*, *Escherichia coli*, and *Salmonella typhi* bacteria. This can be attributed to a scientific basis to support our prediction that this plant is useful as a natural antibiotic.*Prunus cerasus* or sour cherry were also predicted as natural antibiotic candidates in our study. This plant grows in so many countries including Poland, the United States of America, Turkey, Russia, Serbia, Hungary, Iran, Austria, Azerbaijan, Germany, and Indonesia. This plant is usually called cherry *kersen* in Indonesia which is used as a decoration for cakes. It helps in lowering blood pressure, regulating sugar levels, and strengthening our immune system. Research [22] states that it can obstruct the growth of bacteria which justifies our prediction result that this plant is a natural antibiotic.*Borreria hispida*, commonly known as *gempur batu*, is a plant that belongs to the family *rubiaceae* and the genus *Borreria* has been used by the Indonesian people as a medicinal plant, especially to treat kidney diseases. To emphasize the hypothesis of the research results that *Borreria hispida* can be used as a candidate for natural antibiotics, this plant should exhibit the function of prohibiting bacterial growth or killing bacteria. According to [23], the extracts of this plant can be used against *Bacillus subtilis*, *Bacillus cereus*, *Staphylococcus aureus*, *Pseudomonas aeruginosa*, and *Escherichia coli* using the agar disc diffusion method.*Coptis chinensis* is one of the drugs found in traditional Chinese medicine commonly known as *Huanglian*. The extracts of this plant possess strong properties to hinder bacterial growth. Furthermore, it is also used as a medicine for dysentery, cholera, leukemia, diabetes, and lung cancer [24]. Plants produce berberine alkaloids, coptisine, and palmatine which can slow down the growth of *Escherichia coli* [25]. Additionally, referring to the KNApSAck family database, it can be said this plant has biological activity as antibacterial and/or antibiotics.*Cassia alata*, a plant with extreme effectiveness is commonly known as *ketepeng cina* in Indonesia. This plant has several names according to various regions in Indonesia. For example, it is called *kupang* leaf in the Malay area, *ki* manila in the *Sunda* area, *kupang-kupang* in Madura, and *ketepeng cina* in east and central Java. The leaves of this plant are traditionally used to treat scurvy and malaria. According to [26], the contents of *Cassia alata* leaf can inhibit the growth of *Dermathophilus congolensis*, *Staphylococcus aureus*, *Corynebacterium parvum*, *Actinomyces bovis*, and *Clostridium septicum*. This plant has biological activity as antibacterial or antibiotics according to the KNApSAck family database.*Brucea javanica* is commonly known as *buah* makasar or amber *merica* with a bitter taste and is classified as toxic. However, this plant is used as a medicine to prevent dysentery, diarrhea, and malaria. As stated in [27], the potions of its fruits produced a new antibacterial compound for *Streptococcus pyogenes* bacteria where the effective compound is the bitter-tasting alkaloid called brucine. This reference can be utilized as reasoning for predicting this plant as a candidate for natural antibiotics in this study.*Aglaia odorata* or commonly known as *pacar cina* is a plant that has efficacies such as healing bloating, throat, cough, ulcer, and also speeding up of labor. According to [28], stem-derived essential oil from this plant can slow down the growth of Gram-positive and Gram-negative bacteria such as *Bacillus cereus* ATCC 11778, *Staphylococcus aureus* ATCC 25923, *Acinetobacter baumannii* ATCC 19606 and *Escherichia coli* ATCC 25922. Referring to the TCM database, it is explained that this plant can cure abscess disease. Abscess disease is a painful collection of pus, usually emanating from a bacterial infection.*Costus speciosus* is a plant that has a height of about 0.5–3 m with a humid and shady living habit. In Indonesia, this plant has many names such as *pancing*, *pempung tawar*, *poncang-pancing*, *tubu-tubu* and so on. Traditionally this plant is used for various diseases such as kidney disease, stomach ulcer, urinary tract infection, and liver constriction. From [29], we came to know that this plant has several pharmacological activities such as antibacterial, antifungal, anticholinesterase, antioxidant, antihyperglycemic, anti-inflammatory, analgesic, antipyretic, antidiuretic, larvicidal, antistress and estrogenic activity.*Stachytarpheta jamaicensis* or commonly known as pecut kuda, is a wild plant commonly found in Indonesia and has diverse efficacy as per the beliefs of Indonesian people. According to [30], this plant is habitually used to treat digestive, allergic, and respiratory diseases namely asthma, cold, flu, and cough. The plant extracts can be used as an inhibitor for the growth of the following bacteria and fungus: *Bacillus subtilis*, *Escherichia coli*, *Staphylococcus aureus*, *Pseudomonas aeruginosa*, *Proteus vulgaris*, *Klebsiella aerogenes*, *Proteus mirabilis* and *Candida albicans* [31]. In KNApSAck family database it is recorded that this plant has biological activity as antibacterial and/or antibiotics.*Trichosanthes kirilowii* belongs to the *cucurbitaceae* family which has effectiveness against abscess disease according to the TCM database. This abscess disease is generally caused by a bacterial infection and, therefore, it can be concluded that this plant has a direct or indirect relationship in prohibiting bacterial growth. Referring to [32], this plant produces a compound 1-C-(p-Hydroxyphenyl)-Glycerol which can hamper bacterial growth of *Bacillus cereus*, *Escherichia coli*, and *Streptococcus faecalis*.

Out of 14 predicted plants, the following 4 plants can be considered as new natural antibiotics based on the Random Forest model. To the best of our knowledge, we found no articles, journals or online databases that can directly or indirectly mention these plants as antibiotics or inhibiting bacterial growth. Below we discuss some properties of these four plants.

*Prunus armeniaca* L. is a medicinal plant commonly known as apricot and is normally eaten because of its delicious taste. In addition, this plant can also be used as medicine due to properties such as antimicrobial, antimutagenic, inhibiting enzymes, cardioprotective, anti-inflammatory, and antinociceptive. This plant is rich in polysaccharides, polyphenols, fatty acids, sterol derivatives, carotenoids, cyanogenic glucosides, and volatile components that make this plant produce a pleasant aroma [33].*Fritillariae cirrhosae bulbus*, a medicinal plant known as *chuan bei mu* in China, has been used as medicine for a long time for remedies against cough and phlegm. This plant has biological activities such as antitussive, expectorant, analgesic, anticancer, anti-inflammatory, and antioxidative. Moreover, this plant has therapeutic effects on many diseases such as cancer, acute lung injury, chronic obstructive, pulmonary diseases, asthma, Parkinson’s disease, and diabetes [34]. Thus, we assume that it has potential as natural antibiotic for its anti-inflammatory attribute.*Scaphium affinis* is a plant from Indonesia which goes by popular names such as *tempayang* or *semangkuk*. It appears brown and is shaped like *melinjo* seed. As per the traditional belief, this plant can treat diseases such as fever, acute cough, sore throat, hemorrhoids, and increase female fertility.*Pueraria lobata* is one of the plants that has usefulness based in traditional Chinese medicine. A common name for this plant is kudzu in the continent of Asia. This plant is used in the preparation of many foods and cosmetics. In addition, this plant also has potential for biological activities such as antioxidant, antiglycation, skin generation, and melanogenesis inhibitory [35].

## 3. Discussion

In this section, we discuss the labeling of the dataset, the validation of the results, the limitations of this research, and future work that can be continued. An herbal formula that can cure diseases caused by bacteria is assigned to class 1. We performed mapping for each herbal formula in cases such as cough, urethritis, typhoid, and so on and categorized these diseases as caused by bacteria. On the other hand, many diseases such as headaches, indigestion, fatigue, and loss of appetite are not caused by bacteria. This labeling process has many challenges considering the fact that there is no particular database showing whether a disease is caused by bacteria or not. However, the basic knowledge of the medical field can be used to map the class labels accordingly.

The next thing that needs to be discussed is determining a proper machine learning method to model the dataset because the higher model accuracy could give a better result for extracting important features. Using the Random Forest classifier with 10-fold cross-validation, we obtained maximum accuracy, recall, and precision of approximately 91% with a standard deviation of about 1%. Such low standard deviation indicates two things: firstly, the Random Forest classifier performed robustly in the case of modeling the Jamu formula dataset, and secondly, the model with the highest accuracy in a certain fold can be used as the best model to extract important features because it can be concluded that the model is not overfitting.

The extraction process is based on the principle to select the features that are the most important to constructing the trees in Random Forest. The features used are the nodes in the formed trees, and the value of importance is calculated using the package *permutation_importance* available in the *Scikit-learn* library. The importance of each feature is indicated by a numeric value. After filtering and sorting, we selected 14 plants with the highest feature importance values. The results turned out to be quite good because, among these 14 plants, 10 were supported by scientific articles stating that they had been used in killing or inhibiting the growth of bacteria. By further investigating the specific chemical compounds related to the predicted plants we found some supportive evidence. According to the KNApSAcK database, *Coptis chinensis* has several metabolites; one of them is berberine. Berberine metabolite in *Coptis chinensis* plants can increase the antibacterial activity against *Staphylococcus* strain in vitro [36]. *Trichosanthes kirilowii* has several metabolites; one of them is lauric acid. According to the Journal, this metabolite has an antibacterial effect on Gram-positive bacteria [37]. *Stachytarpheta jamaicensis* contains 3-O-Caffeoylquinic acid. Refers to [38], 3-O-Caffeoylquinic acid shows considerable antibacterial activity against *Staphylococcus aureus* and *Escherichia coli*. *Costus speciosus* has several metabolites; one of them is diosgenin. Based on Journal [39], this metabolite has antibacterial activity on *Porphyromonas gingivalis* and *Prevotella intermedia*. *Brucea javanica* has several metabolites one of them is Javanicin. Based on [40], this metabolite has strong antibacterial activity against *Pseudomonas* spp. *Cassia alata* has the metabolite Chrysophanol based on the KNApSAcK database. This metabolite shows substantial antibacterial activity against *E. coli* [41]. *Prunus cerasus* has several metabolites; one of them is chrysin. This metabolite has biological activities such as anticancer, anti-inflammatory, and antiallergic. Derivatives of this metabolite have antibacterial activity against a panel of susceptible and resistant Gram-positive and Gram-negative [42] bacteria.

A limitation of this research is that we could not figure out which specific part of the plants can be used as an antibacterial compound. It can be either from the leaf extracts, fruits, or even from the metabolite content of the plant. The limitation of this work can be used as a theme to continue further study in the future. Additionally, four plants were categorized as newly predicted plants that can be utilized as materials for making natural antibiotics.

## 4. Materials and Methods

The steps executed in this study have been illustrated below (Figure 2). There are five steps: data acquisition, pre-processing, modeling, extraction, and validation.

### 4.1. Data Acquisition

This study used data on herbal formulas from the KNApSAcK database (http://www.knapsackfamily.com/KNApSAcK_Family/), accessed on 30 October 2021 [43]. The research data comprised 465 plants, 3138 Jamu formulas, and 116 diseases that could be cured by Jamu formulas. To perform the prediction task related to antibiotics, 116 diseases were categorized as follows: diseases caused by bacteria (class 1), diseases caused by other microorganisms (class 2), and the rest as class 0.

### 4.2. Pre-Processing

Data preparation includes checking and deleting redundant data, checking for missing values, and deleting Jamu formulas that treat diseases caused by other microorganisms (class 2) to ensure more focus on bacterial diseases. The final dataset is a matrix of Jamu formula versus plants with a column for the class label as shown in Table 5. The value of a cell representing the jth row and kth column is 1, if the jth herbal formula uses the plant corresponding to the kth feature, otherwise it is 0. Class label consists of two values: 0 means the particular Jamu formula does not have efficacy to cure bacterial diseases and 1 means it has.

### 4.3. Modeling

We applied Random Forest classifier. According to preliminary modeling of the dataset, we found that RF is the best model for our dataset. RF is a method that creates a number of classification trees with randomly selected features. Random Forest is a supervised learning method that can determine the class or category of the data. Bootstrap sampling, random feature selection, full-depth decision tree building, and out-of-bag error estimation are the four steps of this method. An illustration of class determination in the RF method can be seen in Figure 3.

Figure 3 explains that an RF classifier forms several decision trees using samples from the dataset/instances. At first, the new data to be classified is tested for all decision trees that have been formed. Then, majority voting is carried out to determine the class label of the new data.

To determine whether a predictive model is good or not, we used several matrices such as accuracy, precision, etc. This study also employs a 10-fold cross-validation method. Results of cross-validation can provide clues to assess the level of overfitting, i.e., the state of the model that fits too well with the data points [44]. Thus, cross-validation can provide a better picture of the model’s ability to perform predictions for new data.

### 4.4. Extraction

The best model obtained is used to determine the important features. The important features are those plants that contribute the most to building the RF model. We used *permutation_importance* in *scikit-learn* library to calculate important features (plants). The inputs of this process are the best prediction model, features data, and class label, and then the output is numeric values for each feature. Furthermore, we filtered and sorted important features based on their values. 

### 4.5. Validation

This study used several approaches to validate plants predicted as natural antibiotics. One of them is tracing directly to scientific journals/articles that describe these plants to be effective for inhibiting bacterial growth. Another is by checking on the open-access databases, which enlist the biological activity properties of plants, such as KNApSAck family database and TCM database (http://www.a-hospital.com/) accessed on 30 October 2021.

## 5. Conclusions

This paper utilizes the formulas of traditional Indonesian medicines (Jamu medicines) to predict plants that can be used as natural antibiotics by machine learning methods. The formulas are classified into two groups, i.e., bacterial and non-bacterial using the Random Forest algorithm. The Random Forest classifier achieves a maximum of 91% accuracy in making predictions and the best classification model is utilized to select 14 important features as natural antibiotic plants. The literature review shows that 10 out of 14 predicted plants are reported to have antibacterial characteristics. These potential natural antibiotic plants are *Clerodendron squamatum*, *Prunus cerasus*, *Borreria hispida*, *Coptis chinensis*, *Cassia alata*, *Brucea javanica*, *Aglaia odorata*, *Costus speciosus*, *Stachytarpheta jamaicensis*, and *Trichosanthes kirilowii*. Moreover, the results of this study can be used as a basis for other studies such as drug discovery, and the discovery of natural antibiotics.

## Figures and Tables

**Figure 1 antibiotics-11-01199-f001:**
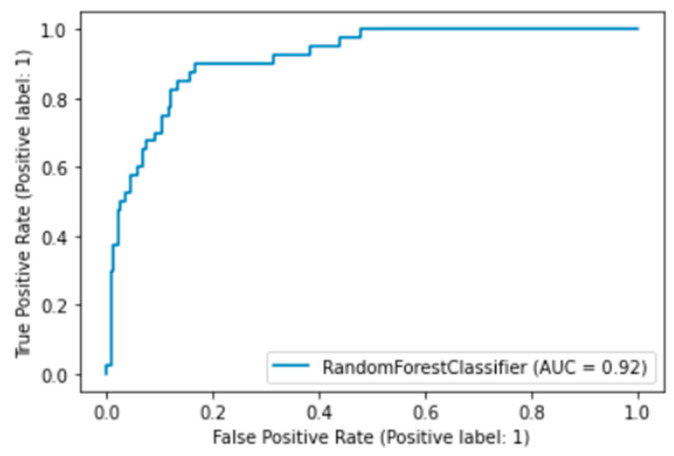
ROC curve for the best model.

**Figure 2 antibiotics-11-01199-f002:**
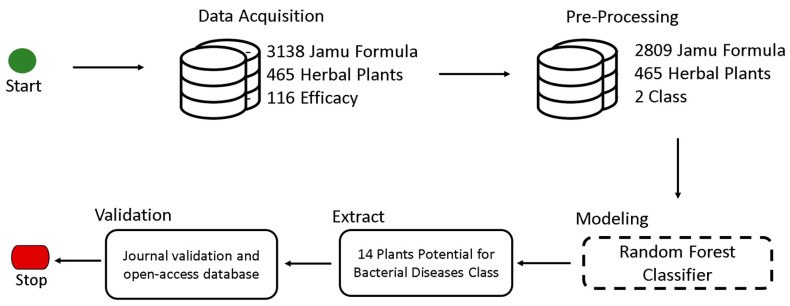
Methodology of research.

**Figure 3 antibiotics-11-01199-f003:**
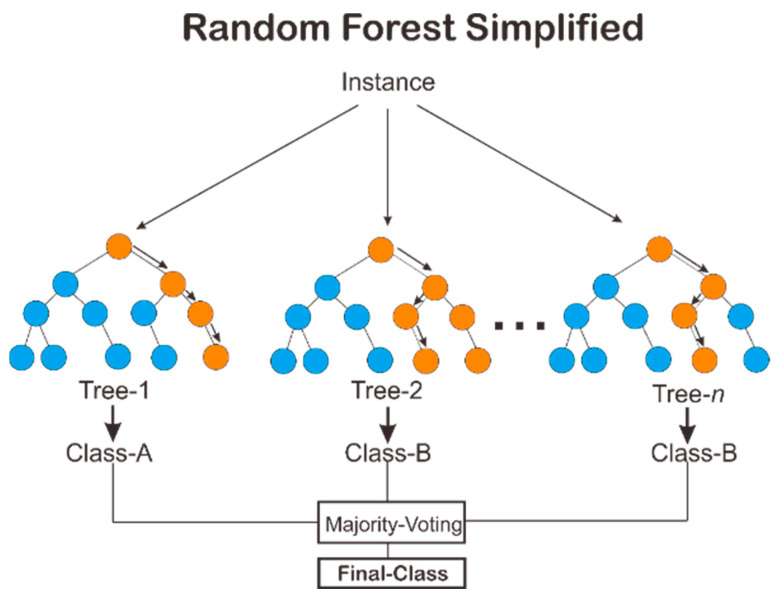
Random Forest classifier.

**Table 1 antibiotics-11-01199-t001:** Preliminary modeling.

Model	Accuracy	Balanced Accuracy	ROC-AUC	F1-Score	Required Time
RandomForestClassifier	0.81	0.79	0.79	0.80	0.75
ExtraTreesClassifier	0.79	0.78	0.78	0.79	0.76
LGBMClassifier	0.78	0.76	0.76	0.77	0.26
BaggingClassifier	0.76	0.75	0.75	0.76	0.50
XGBClassifier	0.77	0.75	0.75	0.77	1.13
DecisionTreeClassifier	0.75	0.75	0.75	0.75	0.13
NuSVC	0.76	0.74	0.74	0.75	2.77
KNeighborsClassifier	0.73	0.73	0.73	0.74	1.68
NearestCentroid	0.73	0.73	0.73	0.73	0.10
AdaBoostClassifier	0.75	0.73	0.73	0.75	0.71
ExtraTreeClassifier	0.73	0.73	0.73	0.73	0.07
LogistciRegression	0.74	0.72	0.72	0.74	0.15
LinearSVC	0.74	0.72	0.72	0.74	1.72
LinearDiscriminantAnalysis	0.74	0.72	0.72	0.73	0.21
BernouliNB	0.73	0.71	0.71	0.72	0.09
SGDClassifier	0.72	0.70	0.70	0.72	0.19

**Table 2 antibiotics-11-01199-t002:** Tuning parameters in Random Forest.

Parameter Name	Parameter Value
n_estimators	200, 400, …, 2000
min_samples_split	2, 5, 10
min_samples_leaf	1, 2, 4
max_features	‘auto’, ‘sqrt’
max_depth	10, 20, …, 110
bootstrap	“True”, “False”

**Table 3 antibiotics-11-01199-t003:** Metrics for dataset using Random Forest classifier.

Fold	Accuracy	Recall	Precision
1	87.90%	87.90%	86.21%
2	89.32%	89.32%	88.36%
3	87.90%	87.90%	86.30%
4	88.26%	88.26%	88.33%
5	91.10%	91.10%	90.54%
6	89.32%	89.32%	88.59%
7	88.26%	88.26%	86.68%
8	87.90%	87.90%	86.15%
9	87.90%	87.90%	86.15%
10	90.00%	90.00%	89.15%
Min	87.90%	87.90%	86.15%
Avg	88.79%	88.79%	87.65%
Std	1.05%	1.05%	1.48%

**Table 5 antibiotics-11-01199-t005:** Representation of dataset in the form of two-dimensional matrix.

Jamu Formula	Plants					Class Label
	P_1_	P_2_	P_3_	…	P_465_	
J_1_	0	0	0	…	0	0
J_2_	0	0	0	…	0	1
J_3_	1	1	1	…	1	0
…	…	…	…	…	…	…
J_2809_	1	1	0	…	0	0

## Data Availability

Dataset is provided on: https://github.com/kamalNasution/Jamu_plants_antibiotics.

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
