# Peer review of "Prediction of Potential Natural Antibiotics Plants Based on Jamu Formula Using Random Forest Classifier"

_antibiotics, 2022, doi:10.3390/antibiotics11091199_

Round 1
Reviewer 1 Report (Previous Reviewer 2)
The authors addressed all my questions very clearly. Now the article is improved a lot as compared to before. I recommend for its publication.
Author Response
Cover letter
Date: 19 August 2022
To
The Editor
MDPI antibiotics Journal
Dear Dr. Editor,
We are submitting our revised paper entitled " Prediction of Potential Natural Antibiotics Plants Based on Jamu Formula Using Random Forest Classifier " to be published in your reputed journal.
First, we heartily thank our reviewers for wise and essential comments, which helped us improve our manuscript's quality substantially. We confirm the following sections in the revised manuscript with appropriate information based on comments.
- Introduction
- Discussion
- Material and methods
This paper reports that 14 potential plants predicted from Jamu formula ingredients become natural antibiotics plants. The result is significant because ten of 14 potential plants have antibacterial properties based on the journal article. Our machine learning model achieved 91% accuracy with a guaranteed robust model from ten-cross fold validation. In this research, we also think the result is useful for other fields of research to develop new natural antibiotics and improve the utilization of Jamu medicine. We believe this manuscript is appropriate for publication in the Antibiotics journal on MDPI because it is suitable for its aim and scope.
At the end of this letter, we are providing a point-by-point answer to the previous comments of the reviewers. We want to thank anonymous reviewers for their insightful suggestions that helped enhance the current paper's quality.
We hope our revised manuscript will be accepted for publication in your reputed journal.
Best regards,
Ahmad Kamal Nasution (e-mail: nasution.ahmad_kamal.my2@is.naist.jp)
Doctoral student
Computational Systems Biology Lab
NAIST, JAPAN
Reviewer 1
Reviewer’s comment:
The authors addressed all my questions very clearly. Now the article is improved a lot as compared to before. I recommend for its publication.
Our reply:
We thank the reviewer and appreciate the comment.

Reviewer 2 Report (New Reviewer)
In this study, a Random Forest (RF) algorithm was employed for the prediction of potential natural antibiotics plants based on Jamu formula. The model obtained is good and robust with the 10-fold cross-validations, the maximum accuracy, recall, and precision as 91.10%, 91.10%, and 90.54%, respectively. As a result, 14 plants were predicted as natural antibiotic candidates. Ten of them have direct or indirect antibacterial activity according to scientific journals. However, there seems no novelty for the method, and there were too little discussion on the specific chemical components with antibiotic activity from the 14 plants. So, we think the manuscript is not suited to Antibiotics.
Author Response
Cover letter
Date: 19 August 2022
To
The Editor
MDPI antibiotics Journal
Dear Dr. Editor,
We are submitting our revised paper entitled " Prediction of Potential Natural Antibiotics Plants Based on Jamu Formula Using Random Forest Classifier " to be published in your reputed journal.
First, we heartily thank our reviewers for wise and essential comments, which helped us improve our manuscript's quality substantially. We confirm the following sections in the revised manuscript with appropriate information based on comments.
- Introduction
- Discussion
- Material and methods
This paper reports that 14 potential plants predicted from Jamu formula ingredients become natural antibiotics plants. The result is significant because ten of 14 potential plants have antibacterial properties based on the journal article. Our machine learning model achieved 91% accuracy with a guaranteed robust model from ten-cross fold validation. In this research, we also think the result is useful for other fields of research to develop new natural antibiotics and improve the utilization of Jamu medicine. We believe this manuscript is appropriate for publication in the Antibiotics journal on MDPI because it is suitable for its aim and scope.
At the end of this letter, we are providing a point-by-point answer to the previous comments of the reviewers. We want to thank anonymous reviewers for their insightful suggestions that helped enhance the current paper's quality.
We hope our revised manuscript will be accepted for publication in your reputed journal.
Best regards,
Ahmad Kamal Nasution (e-mail: nasution.ahmad_kamal.my2@is.naist.jp)
Doctoral student
Computational Systems Biology Lab
NAIST, JAPAN
Reviewer 2
Reviewer’s comment:
In this study, a Random Forest (RF) algorithm was employed for the prediction of potential natural antibiotics plants based on Jamu formula. The model obtained is good and robust with the 10-fold cross-validations, the maximum accuracy, recall, and precision as 91.10%, 91.10%, and 90.54%, respectively. As a result, 14 plants were predicted as natural antibiotic candidates. Ten of them have direct or indirect antibacterial activity according to scientific journals. However, there seems no novelty for the method, and there were too little discussion on the specific chemical components with antibiotic activity from the 14 plants. So, we think the manuscript is not suited to Antibiotics.
Our reply:
We thank the reviewer for the advice and insight. In this paper, we limit the experiment only to plant level for funding natural antibiotic plant candidates. However, we try to add sentences to elaborate on the specific chemical components for seven of the predicted plants.
Updated on manuscript:
Discussion section paragraph 3:
“… By further investigating the specific chemical compounds related to the predicted plants we found some supportive evidences. According to the KNApSAcK database, Coptis chinensis has several metabolites; one of them is Berberine. Berberine metabolite in Coptis chinensis plants can increase the antibacterial activity against Staphylococcus strain in vitro [36]. Trichosanthes kirilowii has several metabolites; one of them is Lauric acid. According to the Journal, this metabolite has an antibacterial effect on Gram-positive bacteria [37]. Stachytarpheta jamaicensis contains 3-O-Caffeoylquinic acid. Refers to [38], 3-O-Caffeoylquinic acid shows considerable antibacterial activity to Staphylococcus aureus and Escherichia coli. Costus speciosus has several metabolites; one of them is Diosgenin. Based on Journal, this metabolite has antibacterial activity on Porphyromonas gingivalis and Prevotella intermedia [39]. Brucea javanica has several metabolites one of them is Javanicin. Based on [40], this metabolite has strong antibacterial activity against Pseudomonas spp. Cassia alata has the metabolite Chrysophanol based on the KNApSAcK database. This metabolite shows substantial antibacterial activity against E. coli [41]. Prunus cerasus has several metabolites; one of them is chrysin. This metabolite has biological activities like anticancer, anti-inflammatory, and antiallergic. Derivatives of this metabolite have antibacterial activity against a panel of susceptible and resistant Gram-positive and Gram-negative [42] bacteria.…. “

Reviewer 3 Report (New Reviewer)
Dear Authors,
Introduction
Which strains of bacteria belong to superbugs?
“This study found eight antibiotic compounds with mostly different structures compered to known antibiotics”…… list the given compounds.
“Besides, high-level species can also produce compounds that inhibit bacterial growth (bacteriostatic)”…. to which high-level species it refers?
Discussion
“Our colleague who is from a medical background carried out this labelling task”. Unnecessary sentence
Material and methods
4.1
“3138 herbal formulas”… to which herbal formulas it refers?
“This mapping was performed by our colleague with the medical background”. Unnecessary sentence
Author Response
Cover letter
Date: 19 August 2022
To
The Editor
MDPI antibiotics Journal
Dear Dr. Editor,
We are submitting our revised paper entitled " Prediction of Potential Natural Antibiotics Plants Based on Jamu Formula Using Random Forest Classifier " to be published in your reputed journal.
First, we heartily thank our reviewers for wise and essential comments, which helped us improve our manuscript's quality substantially. We confirm the following sections in the revised manuscript with appropriate information based on comments.
- Introduction
- Discussion
- Material and methods
This paper reports that 14 potential plants predicted from Jamu formula ingredients become natural antibiotics plants. The result is significant because ten of 14 potential plants have antibacterial properties based on the journal article. Our machine learning model achieved 91% accuracy with a guaranteed robust model from ten-cross fold validation. In this research, we also think the result is useful for other fields of research to develop new natural antibiotics and improve the utilization of Jamu medicine. We believe this manuscript is appropriate for publication in the Antibiotics journal on MDPI because it is suitable for its aim and scope.
At the end of this letter, we are providing a point-by-point answer to the previous comments of the reviewers. We want to thank anonymous reviewers for their insightful suggestions that helped enhance the current paper's quality.
We hope our revised manuscript will be accepted for publication in your reputed journal.
Best regards,
Ahmad Kamal Nasution (e-mail: nasution.ahmad_kamal.my2@is.naist.jp)
Doctoral student
Computational Systems Biology Lab
NAIST, JAPAN
Reviewer 3
Reviewer’s comment:
Introduction
- Which strains of bacteria belong to superbugs?
- “This study found eight antibiotic compounds with mostly different structures compered to known antibiotics”…… list the given compounds.
- “Besides, high-level species can also produce compounds that inhibit bacterial growth (bacteriostatic)”…. to which high-level species it refers?
Our reply:
We thank the reviewer for the comment and question. As the reviewer commented, Point 1, we updated the manuscript to make the introduction clear. Point 2, we added the eight compounds using ZINC15 database id according to the corresponding reference. Point 3, high-level species refers to plants.
Updated on manuscript:
- Introduction paragraph 2:
“Superbugs are bacteria that can fight drugs or antibiotics e.g. Staphylococcus aureus resistant to Methicilin [3] … “
- Introduction paragraph 4:
“… This study used a deep neural network to predict molecules with an antibacterial activity using various database sources such as drug repurposing hub and ZINC15. This study found eight antibiotic compounds (ZINC000098210492, ZINC000001735150, ZINC000225434673, ZINC000004481415, ZINC000019771150, ZINC000004623615, ZINC000238901709, and ZINC000100032716) … “
- Introduction paragraph 5:
“… Antibiotics themselves are usually created from microorganisms that are toxic to other microorganisms (bacteria). Besides, plants can also produce compounds that inhibit bacterial growth (bacteriostatic)… “
Reviewer’s comment:
Discussion
“Our colleague who is from a medical background carried out this labelling task”. Unnecessary sentence
Our reply:
We thank the reviewer for the comment. We have removed these sentences from the revised manuscript.
Reviewer’s comment:
Material and methods
4.1
- “3138 herbal formulas”… to which herbal formulas it refers?
- “This mapping was performed by our colleague with the medical background”. Unnecessary sentence
Our reply:
We thank the reviewer for the comment and advice.
Point 1. We clarify that 3138 herbal formulas in this section refer to the Jamu formula, so we have updated the manuscript.
Point 2, We have removed these sentences from the revised manuscript.
Updated on manuscript:
Material and Methods. 4.1
“ This study used data on herbal formulas from the KNApSAcK database (http://www.knapsackfamily.com/KNApSAcK_Family/) [37]. The research data comprised 465 plants, 3138 Jamu formulas, and 116 diseases that could be cured by Jamu formulas. To perform the prediction task related to antibiotics, 116 diseases were categorized as follows: diseases caused by bacteria (class 1), diseases caused by other microorganisms (class 2), and the rest as class 0. “

Round 2
Reviewer 2 Report (New Reviewer)
The responses of the authors are acceptable. The revised manuscript is ready for publication as it is.
Reviewer 3 Report (New Reviewer)
I can recommend publication of the manuscript in its present state
This manuscript is a resubmission of an earlier submission. The following is a list of the peer review reports and author responses from that submission.
Round 1
Reviewer 1 Report
This manuscript presents the use of a random forest classifier to predict the antibiotic activity of natural products. The manuscript is technically sound, however, it lacks key literature on the topic of antibiotics and antibiotic resistance. The authors should also critically read the manuscript because in its present state it contains several typos and imperfections.
Specific comments:
1) Tile: change machine learning approach to random forest classifier
2) Abstract: please explain to the general audience what the data about your accuracy, recall, precision and standard deviations imply. At the moment this part is very unclear.
3) Introduction:
- please remove the sentence “In addition, it has been speculated that up to 75% 37 of the entire plant species grow in Indonesia” since this statement is not backed up by scientific evidence.
- please remove the sentence “The bacterium is a single-cell organism that has a size smaller than 1 micron and is 39 divided into four divisions, namely, Gracilicutes (Division 1), Firmicutes (Division 2), 40 Tenericutes (Division 3), and Mendosicutes (Division 4)” this is trivial
- please expand you introduction to antimicrobial treatment and antibiotic resistance that is currently too brief and lacks several important recent studies, for example about persisters and viable but non culturable cells:
BMC biology 15, 121 (2017)
PLoS pathogens 17, e1009194 (2021)
eLife 11, e74062 (2022)
mBio 12, e00909, (2021)
mBio 12, e00703, (2021)
4) Results:
- please explain what is the KNApSAcK database
- please comment on your results presented in table 1, at the moment it is not clear what it is presented, e.g. what is the best model and why?
Author Response
Date: 17 June 2022
To
The Editor
MDPI antibiotics Journal
Dear Dr. Editor,
We are submitting the revised version of our manuscript titled " Prediction of Potential Natural Antibiotics Plants Based on Jamu Formula Using Random Forest Classifier " to be published in your reputed journal.
First, we heartily thank our reviewers for wise and important comments which really helped us to substantially improve the quality of our manuscript. Based on your comments we updated the following sections in the revised manuscript with appropriate information.
- Title
- Abstract
- Introduction
- Result
- References
- Writing format
This paper reports that 14 potential plants predicted from Jamu formula ingredients are important natural antibiotic plants. The result is significant because 10 out of 14 potential plants are reported to have antibacterial properties based on the journal articles. Our machine learning model achieved 91% accuracy with a guaranteed robust model from ten-cross fold validations. In this research, we also think the result is useful for other research field, e.g., to develop new natural antibiotics and improve the utilization of Jamu medicine. We believe that this manuscript is appropriate for publication in the Antibiotics journal on MDPI because it is suitable for the journal’s aim and scope.
At the end of this letter, we are providing point by point answer to the comments of the reviewer 1. We would like to address our gratitude to anonymous reviewers for their insightful suggestions that helped to enhance the quality of the current paper.
We hope our revised manuscript will be accepted for publication in your well-known journal.
Best regards,
Ahmad Kamal Nasution
Doctoral student
Computational Systems Biology Lab
NAIST, JAPAN
e-mail: Nasution.ahmad_kamal.my2@is.naist.jp
Reviewer 1
Reviewer’s comment:
Title: change machine learning approach to random forest classifier
Our reply:
We agree to the comment because we only used the Random Forest classifier model to extract new candidate antibiotic plants based on Jamu fprmulas.
Reviewer’s comments:
Abstract: please explain to the general audience what the data about your accuracy, recall, precision and standard deviations imply. At the moment this part is very unclear
Our reply:
We have explained what the accuracy, recall, precision, and standard deviation imply in the revised manuscript.
Reviewer’s comment:
Introduction
- please remove the sentence “In addition, it has been speculated that up to 75% 37 of the entire plant species grow in Indonesia” since this statement is not backed up by scientific evidence.
- please remove the sentence “The bacterium is a single-cell organism that has a size smaller than 1 micron and is 39 divided into four divisions, namely, Gracilicutes (Division 1), Firmicutes (Division 2), 40 Tenericutes (Division 3), and Mendosicutes (Division 4)” this is trivial
- please expand you introduction to antimicrobial treatment and antibiotic resistance that is currently too brief and lacks several important recent studies, for example about persisters and viable but non culturable cells:
Our reply:
We thank to reviewer for this advice. We agree and removed those sentences and expanded the Introduction using several references (BMC biology 15, 121 (2017), PLoS pathogens 17, e1009194 (2021) eLife 11, e74062 (2022) mBio 12, e00909, (2021) mBio 12, e00703, (2021)) and added some other relevant references to support our Introduction.
Reviewer’s comment:
Result
- please explain what is the KNApSAcK database
- please comment on your results presented in table 1, at the moment it is not clear what it is presented, e.g. what is the best model and why?
Our reply:
We added some explanations about the KNApSAcK database (Afendi et al., 2012). We also elaborated explanation of Table 1. We have chosen the Random Forest classifier because that model produced higher accuracy, balanced accuracy, and F1-Score compared to other methods.

Reviewer 2 Report
Dear authors after reading out your paper, I come out with these comments.
1. Initially your introduction section is very small and not explanatory, and the references are not up to date in the introduction even some references are from 1999, and 2003. The introduction section paragraphs 2and 3 are very simple. Kindly improve it. Only 4 references have been cited in the intro section. Kindly update and ad these updated references to ur article PMID: 33572497 , PMID: 31081776, Molecules 2022, 27(11), 3630.
2. Table 4. the name of all plants should be written in the italic form and also the name of the genus throughout the paper write it in italic form.
3. Line 132, 133, 157, 166-168, 173, 182-183, 201-203, the name of the bacteria Staphylococcus aureus, and others all should be written in italic. Kindly revise it throughout the article.
4. Section 2.4. I strongly refer to organize one table and mention it clearly about plant name, covering the spectrum diseases like anti-bacterial/anti-fungal and then write the name of those bacteria like (S.aureus) or (C.albicans), then the origin from where this plant was taken like Indonesia, or china. so that it would be easy for readers to read.
5. The references are not prepared according to journal style. some are with et. al, and some references are totally missing the data like reference 20. kindly properly arrange your article.
Author Response
Date: 17 June 2022
To
The Editor
MDPI antibiotics Journal
Dear Dr. Editor,
We are submitting the revised version of our manuscript titled " Prediction of Potential Natural Antibiotics Plants Based on Jamu Formula Using Random Forest Classifier " to be published in your reputed journal.
First, we heartily thank our reviewers for wise and important comments which really helped us to substantially improve the quality of our manuscript. Based on your comments we updated the following sections in the revised manuscript with appropriate information.
- Title
- Abstract
- Introduction
- Result
- References
- Writing format
This paper reports that 14 potential plants predicted from Jamu formula ingredients are important natural antibiotic plants. The result is significant because 10 out of 14 potential plants are reported to have antibacterial properties based on the journal articles. Our machine learning model achieved 91% accuracy with a guaranteed robust model from ten-cross fold validations. In this research, we also think the result is useful for other research field, e.g., to develop new natural antibiotics and improve the utilization of Jamu medicine. We believe that this manuscript is appropriate for publication in the Antibiotics journal on MDPI because it is suitable for the journal’s aim and scope.
At the end of this letter, we are providing point by point answer to the comments of the reviewer 2. We would like to address our gratitude to anonymous reviewers for their insightful suggestions that helped to enhance the quality of the current paper.
We hope our revised manuscript will be accepted for publication in your well-known journal.
Best regards,
Ahmad Kamal Nasution
Doctoral student
Computational Systems Biology Lab
NAIST, JAPAN
e-mail: Nasution.ahmad_kamal.my2@is.naist.jp
Reviewer 2
Reviewer’s comments:
Initially your introduction section is very small and not explanatory, and the references are not up to date in the introduction even some references are from 1999, and 2003. The introduction section paragraphs 2and 3 are very simple. Kindly improve it. Only 4 references have been cited in the intro section. Kindly update and ad these updated references to ur article PMID: 33572497, PMID: 31081776, Molecules 2022, 27(11), 3630.
Our reply:
We thank the anonymous reviewer for this advice, we agree that the introduction in the previous version of the manuscript was too short and insufficient. So, we expanded the introduction in the revised manuscript and added some new references including the following: PMID: 33572497, PMID: 31081776, Molecules 2022, 27(11), 3630.
Reviewer’s comments:
- Table 4. the name of all plants should be written in the italic form and also the name of the genus throughout the paper write it in italic form.
- Line 132, 133, 157, 166-168, 173, 182-183, 201-203, the name of the bacteria Staphylococcus aureus, and others all should be written in italic. Kindly revise it throughout the article.
Our reply:
We thank the reviewer. We addressed this comment in the revised submission.
Reviewer’s comment:
Section 2.4. I strongly refer to organize one table and mention it clearly about plant name, covering the spectrum diseases like anti-bacterial/anti-fungal and then write the name of those bacteria like (S.aureus) or (C.albicans), then the origin from where this plant was taken like Indonesia, or china. so that it would be easy for readers to read.
Our reply:
We Thank you for the advice; we agreed and added a table containing the summary of our investigation for ten predicted plants, which could be validated by published literatures. The information contains the plants' names, origins/place avaliable, and properties/effectiveness.
Reviewer’s comment:
The references are not prepared according to journal style. some are with et. al, and some references are totally missing the data like reference 20. kindly properly arrange your article.
Our reply:
We apologize for this mistake. We have revised the references according to the MDPI antibiotics style.

Round 2
Reviewer 1 Report
The authors have addressed the issues raised and the manuscript is now ready for publication
Reviewer 2 Report
The authors mentioned in their response letter that they already answered all the questions which actually they didn't. The quick revision always let many errors unsolved. They improved a little as compared to its initial draft but still, there are a lot of gaps and drawbacks in the article which needed to be carefully revised.
1. In my previous comments the comment no 1, i asked the authors to update references with latest references and remove the old references, the authors only added some but still there are a lot of old references of almost 20 years previous references. Besides, i mentioned authors to up-to-date their article with the three latest references but the authors only cited 1 article in the paper, and in their response letter the authors mentioned that they have cited all three articles in the paper, kindly show me the other two articles where they are cited. The authors should initially respect the ethics of the papers and reviewers, if the authors can address some points they can write that they can address if they cannot they should also mention the reason why they cannot.
2. The authors should use the scientific writing style to write some words e.g in table 4 (Places available), this word should be changed with the proper wording like "habitat" and the word "plant properties" what the authors mean by plant properties? what they mentioned in the article is "spectrum", not properties. or they can write "pharmacological activities". Kindly change it.
3.Table 4. section 2 ; the authors mentioned "bacterial activity". What do you mean by bacterial activity? it should be anti-bacterial activity.
4. Table 4. Kindly remove this word "inhibit bacteria". the heading of the title should be spectrum or pharmacological activity and then directly write the name of bacteria or fungi.
5. In my previous comments point 4. I asked authors to organize the plants and especially microbes and characterize them as anti-bacterial and anti-fungal, but the authors didnt do any changes in that. Table 4. "plant name, Stachytarpheta jamaicensis, in plant properties the authors mentioned inhibit bacteria and they have written the name of "candida albicans" which i alreday mentiond in my previous comments that C.albicans is fungi not bacteria, but still it is written bacteria in the table and in text too.
6. In my previous comments i asked authors to carefullyc heck there refernces, but the refernces are agin toll old and bibiliogrpahic data is missing again for refernce no 32.
7. The article still have a lot of language errors, typo and grmmatical mistakes which need to be properly addressed.